# Intervening to prevent a suicide in a public place: a qualitative study of effective interventions by lay people

Christabel Owens ![ORCID] ,[1] Jane Derges,[1] Charles Abraham[1,2]

[1]University of Exeter Medical School, Exeter, UK
[2]School of Psychological Sciences, University of Melbourne, Melbourne, Victoria, Australia

**Correspondence to**
Christabel Owens;
c.v.owens@exeter.ac.uk

## ABSTRACT

**Objectives** Many suicides take place in public locations, usually involving jumping from high places or use of transport networks. Previous research has largely focused on the effectiveness of physical barriers at frequently used locations. There have been no studies of human intervention to prevent suicides in public places. The aim of this study was to identify the core components of an effective intervention by a member of the public.

**Design and methods** We conducted in-depth qualitative interviews with people who have either been prevented by a stranger from taking their own life in a public location (n=12) or intervened to prevent a stranger from taking their own life in a public location (n=21). Collectively, the two groups narrated 50 incidents of suicide rescue. We analysed interview transcripts using inductive thematic analysis.

**Results** Suicidal people typically displayed no visible distress, describing themselves as being dissociated or 'in a bubble'. Intervention was seen to involve three main tasks: 'bursting the bubble' (reconnecting with self, others and everyday world); moving to a safer location, and summoning help. We show how interveners accomplished these tasks in a range of ways, using both verbal and non-verbal communication and different degrees of restraint.

**Conclusions** This is the first empirical study to examine the role of passing strangers in preventing suicides in public places. It shows that no specialist skills are needed. Interveners were ordinary people, distinguished only by a high level of social awareness, combined with a readiness for social action. The findings also suggest that people do not need a script and should not be afraid of saying 'the wrong thing.' What interveners said was much less important than how they made the suicidal person feel, namely safe, connected and validated ('I matter'). Interveners did this simply by being themselves, responding with authenticity, calmness and compassion. Members of the public need to be encouraged to recognise and reach out to those who may be at risk of suicide in public locations, but should be prepared for a prolonged and intense encounter that may leave them with troubling emotions.

## INTRODUCTION

Suicide accounts for around 6000 UK deaths each year, up to one-third of which occur in the public locations.[1] These may be more easily preventable than those that occur

## Strengths and limitations of this study

► This is the first empirical study to examine the role of passing strangers in preventing suicides.
► We included the perspectives of both interveners and those whose lives had been saved by the intervention of a stranger and found a very high level of agreement between them as to what worked, giving confidence in our conclusions about effective interventions.
► Although we did not specify that interventions must have been successful, our intervener group did not include any individuals who had tried unsuccessfully to avert a suicide, so we are unable to draw any conclusions about ineffective interventions.
► We allowed a wide range in the length of time between incident and interview (from less than 1 year to more than 10 years), extending our original upper limit of 5 years in some cases, after careful screening to assess the quality of recall; all participants were able to give extremely detailed accounts, which we had no reason to doubt.
► Our model of effective suicide crisis intervention may be culturally specific and may not be suited to other countries, particularly those in East Asia that have much higher incidence of suicides in public places.

in the privacy of the home, because of the potential for a chance passer-by to make a last-minute intervention. There is compelling anecdotal evidence that lives can be saved by passing strangers acting on the spur of the moment. 'The Stranger on the Bridge' was an award-winning documentary, broadcast on UK television in 2015, which told the story of one such intervention and captured global media attention.[2]

Preventing suicides in public places occupies a key place in England's national suicide prevention strategy.[3] In 2015, Public Health England published guidance for local authorities to help them identify sites and structures that might be used for suicides and take action to prevent deaths at those locations.[4] These typically include sites that provide opportunity for jumping from a height (bridges, high

buildings and cliffs) and sections of the road and railway networks. Previous research has largely focused on the effectiveness of installing physical barriers at such locations to restrict access to means.[5–7] There are no studies of human intervention, and no evidence-based guidance for members of the public. The literature contains only one published article on the topic of what a potential helper might say in a last-minute crisis intervention.[8] Based on the authors' clinical experience rather than research evidence, it provides a suggested script that is long, verbose and contains no clear messages for the general public; nor has it been tested. The World Health Organization (WHO) has published guidance for emergency services personnel responding to suicidal crises,[9] but the advice is very general and does not differentiate between crises occurring in private and in public places. There is an extensive literature on police crisis negotiation, some of which deals with suicide intervention,[10–12] but this is either too general, in the manner of the WHO document, or too specialised, describing 'tricks of the trade' that could not be safely recommended to lay people.

We were not interested in the part played by health professionals or members of the emergency services, whose jobs include suicide crisis intervention, but in ordinary citizens who happen to be passing by. The aim of this study was to identify the core components of an effective intervention by a stranger in a public place.

## METHODS
### Design
We conducted a qualitative interview study in order to understand intervention by a passing stranger from the perspective of both parties.

### Participants and recruitment
Participants were eligible for inclusion if they were aged 18 or over and had experience of either of the following:
► being stopped from taking their own life in a public location by a passing stranger (survivors);
► trying to stop someone they did not know from taking their own life in a public location (interveners). This group included members of the general public and members of staff in non-health agencies whose work at public locations brings them into contact with suicidal individuals, such as railway staff.

A public location was defined as anywhere outside their own or someone else's home. For both groups, we placed advertisements on the websites of mental health and suicide prevention charities and used the social media accounts (Facebook and Twitter) of non-academic partners to invite people with relevant experience to come forward. Interveners were also sought via relevant agencies, including Network Rail, Highways England and bridge authorities. We continued recruiting until each group contained a reasonable degree of diversity in respect of age, gender and ethnicity and we were confident that no new insights were emerging (data saturation).

### Interviews
We conducted in-depth interviews with both groups, beginning with an invitation to give an uninterrupted account: 'Please tell me in your own words as much as you can remember about [the event in question].' This free narrative approach affords insight into what is uppermost in the participant's mind.[13–15] It also allows them to disclose information at their own pace, thereby increasing their sense of safety when talking about difficult or sensitive subjects.[16 17] During the narrative phase, the interviewer (JD, an experienced qualitative researcher) assumed the role of active listener. She then proceeded to ask open-ended questions, following up any points requiring clarification and probing for further detail where necessary.[18] Separate interview guides for each group were used to ensure that all topics of interest were covered and that safety protocols were followed (available as online supplementary files). Interveners reflected in detail on what they said/did and how key decisions were made. Survivors described what the intervener said/did and how it affected them, allowing us to examine mechanisms of action.

All interviews with survivors were conducted face to face, mostly in participants' homes, and lasted 1–2 hours. Interviews with interveners were conducted face to face (n=15), by telephone (n=4) or Skype (n=2). All participants gave written consent. Interviews were audio recorded, transcribed verbatim and anonymised.

### Analysis
Data collection and analysis were carried out concurrently to allow for progressive focusing of interviews and exploration of emerging ideas. Two investigators (CO and JD) independently read the first three transcripts from each group (n=6) before meeting to discuss candidate themes and analytic strategy. They then worked line-by-line through six further transcripts, noting and naming specific features of the situation, including: characteristics of the actors; discrete intervention elements (verbal and non-verbal) and their effects on the suicidal person; constraints, complicating factors and tipping points. These were used to construct a list of categories, with definitions, that were used for coding the entire data set. Coding, sorting and retrieval were carried out using NVivo software. We then carefully examined all the coded material, exploring each category and its relationship to others, visually mapping intervention elements, and working via a series of iterations and data checks towards our final interpretive model.

### Patient and public involvement
We designed the study in consultation with people with relevant lived experience, representatives from suicide prevention charities and the transport industry. They were instrumental in recruiting participants and subsequently commenting on findings and interpretation. We will continue to work with them as we start to frame key public education messages. A lay summary of results was

sent to participants, who were also invited to help with future message development work.

## RESULTS
### Characteristics of the sample

We recruited 12 people who had been stopped from taking their own life in a public place (survivors): nine female and three male, with ages ranging from 19 to 64. Four of them had been stopped on more than one occasion (one twice and three on three occasions), giving a total of 19 accounts of successful intervention.

The intervener group contained 21 individuals: 9 male and 12 female, also ranging in age from 19 to 64. This number was made up of 13 members of the general public, 6 railway workers (two of whom were off-duty at the time) and two highways officers. The railway staff were all Mobile Operations Managers, whose role is to deal with a range of incidents on the railway network, including threatened suicides and fatalities. They differed from members of the general public insofar as: (1) they were often acting on a referral, so had some time to prepare; (2) five out of the six had received basic suicide intervention training from Samaritans; (3) they had responsibilities beyond caring for the suicidal person, including restoring train services as quickly as possible, and (4) they had organisational arrangements in place for managing such incidents, including direct links to the British Transport Police. Our rational for including them was that they were responding as lay people (that is, not as health professionals or emergency services personnel), to a threat of suicide by a stranger in a public location.

Between them, the intervener group had conducted 31 interventions, all successful. Most of the railway workers and one of the highways officers had intervened successfully on more than one occasion. Three members of the public had conducted more than one rescue. One participant was included in both groups because, having been stopped from taking her own life in a public place, she went on to conduct two successful interventions. The members of the public included students, teachers/lecturers, youth leaders, church and charity workers, civil servants, office workers and an actor. Several of them had lived experience of mental health problems. They recounted how, during the intervention, they drew on a miscellany of skills and scraps of knowledge that they had acquired in other spheres, including work and leisure pursuits. One used the metaphor of an 'experience bag' that she carried around with her and had raided for ideas and possible strategies to use while struggling on two different occasions to keep a suicidal person safe.

Of the 50 interventions recounted (19 by survivors and 31 by interveners), 28 occurred on the rail and road networks (including bridges over road or railway). A further 18 involved other types of bridge or high places. Characteristics of participants and settings are shown in table 1.

| Table 1 | Participants and settings | |
|---|---|---|
| | **Survivor group (n=12)** | **Intervener group (n=21)** |
| Age range | 19–64 | 19–64 |
| Gender | | |
| Female | 9 | 12 |
| Male | 3 | 9 |
| Ethnicity | | |
| White British/European | 10 | 21 |
| Asian/Asian British | 2 | - |
| Black/Black British | - | - |
| Other ethnic groups | - | - |
| Mixed/multiple ethnicity | - | - |
| Location | (no of incidents =19) | (no of incidents =31) |
| Rail/underground/bridge over railway | 4 | 14 |
| Road network/bridge over road | 2 | 8 |
| River/bridge over river | 7 | 4 |
| High building | 3 | 0 |
| Cliffs | 2 | 3 |
| Other | 1 | 2 |
| Time since incident | (n=19) | (n=31) |
| 1 year or less | 7 | 7 |
| 2–5 years | 8 | 10 |
| 6–10 years | 2 | 12 |
| More than 10 years | 2 | 2 |

We present our findings under the following main headings: recognition; three intervention tasks, and endings and aftermath.

### Recognition

The prerequisite for intervention is recognition. Interveners recognised the person as being at risk or in need of help primarily by their location. The person either appeared to have placed themselves in immediate physical danger or was judged to be out of place, giving rise to curiosity or concern:

Intervener (2.06): There was a person standing there in a hoodie, sort of facing the railway and looking over the side … and I took a couple of steps and thought, Hang on, what's she doing there? I did literally stop and think, Hang on, this is odd.

Survivors also acknowledged the oddness of their positioning:

Survivor (1.01): I went very close to the edge of the platform and sat with my feet over the edge … quite hunched up, possibly my hands over my ears and my

eyes closed. I mean, it's not generally a position people take when they're waiting for a train.

There were few clear behavioural clues and rarely anything in their mode of dress or appearance to mark them out. What was striking in interveners' accounts of what they saw was the absence of visible emotion. They described the suicidal person as looking 'vacant', 'glazed over', 'zoned out' or 'as if no-one was at home', but not distressed. This was corroborated in the accounts of survivors, who described themselves as having moved into a space beyond emotion: 'a weird sort of surreal, unfeeling state' in which they were 'completely numb', 'frozen' or 'dissociated', cut-off from themselves, others and the everyday world, to the point where they believed they were invisible. They were not inviting rescue:

> Survivor (1.11): It didn't occur to me that people would stop and take notice. I was very much of this mindset that I was invisible and no-one would see me.

This state of dissociation was described as being 'in a bubble.'

Interveners, on the other hand, appear to have been characterised by a state of openness to what was going on around them and an attunement to human distress even when it was not manifest. Some reflected that this had been acquired from parents with a strong sense of social justice or through experience in particular settings, such as working as a bar manager: 'They teach you to spot trouble before it happens' (Intervener 2.15). This participant had been walking across a bridge with his partner when they encountered a person preparing to take her own life. He knew immediately that something was amiss, while his partner was unaware of anything until he started to intervene.

## Three intervention tasks

We identified three main tasks involved in intervention:
1. 'Bursting the bubble'
2. Moving to a safer location
3. Summoning help.

The three tasks are interconnected, as shown in figure 1 and described below. Interventions were typically complex and interveners' accounts, particularly those of general public members who did not have organisational structures around them, show them to be frantically multitasking during the course of the encounter, trying to think ahead like chess players and plan several possible moves. Occasionally, they managed to achieve several purposes through a single speech act or gesture.

The order in which they approached the tasks and the manner of intervention was influenced by a number of factors, including: the immediacy of danger; clarity of suicidal intent; the capacity of the suicidal person to interact verbally; the intervener's personality, and whether they were in a position of authority and responsibility. Interventions and their constituent parts fell on a spectrum according to the degree of control assumed by the intervener (figure 2). At one end were interventions involving forcible restraint and authoritative language, often necessitated by the perceived urgency of the situation. At the opposite end, some interveners were acutely mindful of the suicidal person's right to self-determination and need to feel in control of their own destiny. Accordingly, they asked permission before taking any action and used invitations and appeals, calling on the person to rescue themselves and allowing them to keep all their options open:

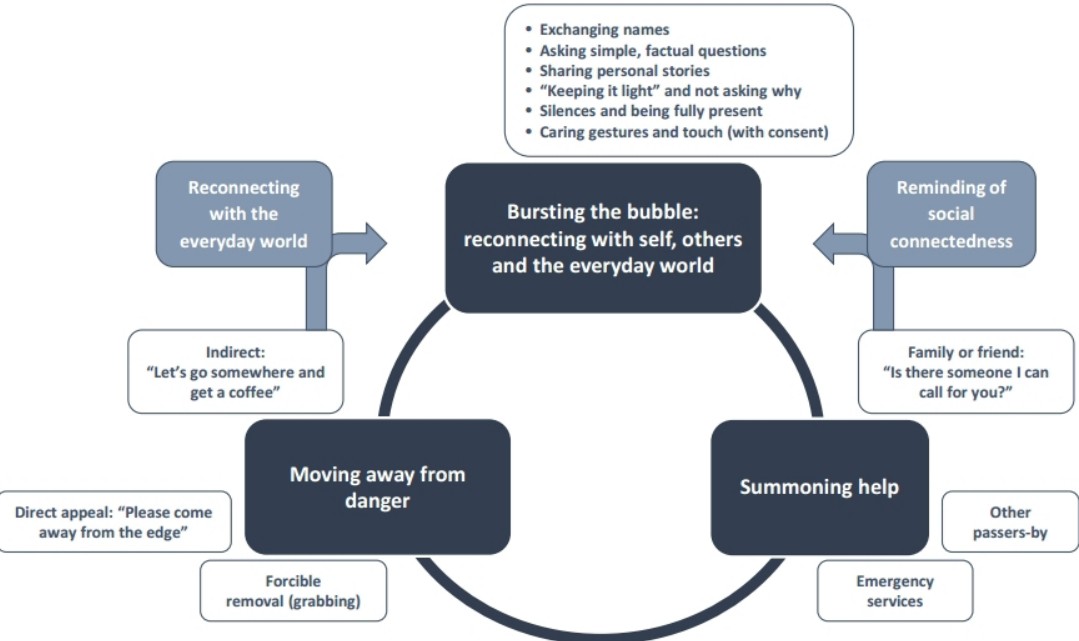

**Figure 1** Main intervention tasks, components and interconnections.

**Taking control:**
physical restraint;
commanding language

**Giving control:**
asking permission;
inviting self-rescue

**Figure 2** Control/agency spectrum.

Survivor (1.05): He came up behind and he said, Look, I don't want to stop you from doing anything but I'd like it if you could come over the barrier … Just step on over, it's OK … He said, I'm not going to take you away from here, you can stay here if you want … It was like, he's not forcing me to do anything. I can still do it … and that was important for me.

### 1) 'Bursting the bubble'

Both interveners and survivors used the metaphor of the bubble and described intervention in terms of bringing the suicidal person out of their trance-like, dissociated state and back into the present moment, reconnecting them with themselves, others and the everyday world. This emotional labour formed the heart of most of the interventions we studied. It was predominantly verbal in nature, although embodied communications (the conveying of a message using bodily activity alone) sometimes also played a key part.

The work began with the intervener establishing their presence. Most approached cautiously and started with some kind of question, such as:

Are you OK?

Can I ask what you're doing there?

You seem to be in a spot of trouble. Can I help?

These apparently simple enquiries did not always elicit a straightforward answer. The intervener still needed to exercise personal judgement, and quickly, as in the following example:

Survivor (1.06): A woman turned up … and she said, Are you OK? I said, I'm fine, thank you … And she said, Well you don't seem fine. I said, I'm just looking at the view. I was angry that she was meddling. It wasn't her business. I was perfectly fine on my own. She said, Mmm, do you want to climb back over and we can both look at the view? And I said, No.

Other interveners were more assertive in their approach, using some degree of verbal or physical restraint:

Survivor (1.07): A woman came over … She put her hand on my arm … and she said, You don't need to do this. It's OK, nothing's that bad.

Having approached the person and made an opening gambit, interveners then found themselves committed to remaining present for as long as it took, either to accomplish a handover to emergency services or to be sure that the person was no longer a risk to themselves. In some cases, this took several hours. During that time, they largely improvised, trusting their instincts, drawing on their personal resources, and trying to show warmth and compassion in whatever way they could. The following elements emerged as important in the accounts of both groups.

Exchanging names: Names were not always exchanged, but when they were this was seen as having played an important part in 'bursting the bubble' and personalising the encounter. By giving their name, interveners demonstrated that they were authentic or 'real', which could have an anchoring effect:

Survivor (1.06): I remember her name. It's the one thing that I kept saying to myself … It was something to hold onto, more tangibly than like a physical thing, I think.

Asking simple, factual questions: Interveners went on to ask simple questions in the usual manner of 'getting to know you', enquiring where the person lived, whether they worked or were studying and so on. This served multiple purposes, beyond establishing the facts. It gave the suicidal person something to focus on, drawing their attention away from imminent death and helping them to reconnect with themselves and their normal world. It also served gradually to build rapport between suicidal person and intervener. One participant highlighted the importance of persisting, if at first the suicidal person is unable to answer:

Survivor (1.01): It did take quite a lot of asking … She kept repeating the same questions … When she asked them the first time I wasn't in a place where I could answer, but as I sort of came back into reality a bit, rather than the world inside my head … You start to feel a bit less disconnected … almost like it's foggy and then the fog lifts a bit, enough for you to see that there is something beyond the fog, like if the fog is suicide.

Interveners also enquired about family members and friends, and if there was someone who could be contacted (see the Summoning help section).

Sharing lived experience and personal stories: As in any normal conversation, interveners reciprocated by sharing information about themselves. When interveners had their own lived experience of mental health problems, participants in both groups believed it was useful to share this in order to establish credibility and trust and to provide hope of recovery. At the same time, interveners were careful to avoid focusing attention on themselves or presuming to know what the suicidal person was going through.

 

'Keeping it light', and not asking why: Participants in both groups expressed a strong belief that this was not the time or place to probe the reasons for wanting to die. Those in the survivor group who had been asked what had brought them to the point of suicide reported that they were unable to talk about it, while others said they were glad they had not been asked. As they emerged from the 'bubble', their predominant feelings were those of ambivalence about being prevented from carrying out their plan, embarrassment and exhaustion, and their immediate need was simply for the comforting presence of another human being, not an intellectually demanding 'talking therapy' session that kept them focused on the distress:

> Survivor (1.06): She started saying, What's happened? Why do you feel like this? And I didn't say anything. I couldn't give her an answer. It was too complicated.

> Survivor (1.11): She didn't ask why… I don't think I would have been able to answer her anyway. I would have probably bolted.

Some interveners did ask what had brought the person to this point and some, but not all, received a response. Others reported that they had shied away from 'getting in too deep' or sensed that it was not appropriate to do so, believing that exploring the precipitating events or stressors could wait until later and that making the person dwell on their problems at this point would not be useful:

> Intervener (2.13): I realised I didn't need to go into the why, didn't need to establish any of that stuff.

Nor was it considered helpful for a stranger to present a suicidal person with reasons for living. This was likely to be regarded as presumptuous and to provoke anger, especially when the intervener lacked lived experience of depression or suicidality.

Instead of dwelling on distress, interveners tried to keep the conversation 'light' and focused on normal, everyday things, a number of them admitting that they were simply trying to buy time or alleviate their own discomfort. Survivors affirmed that this was nonetheless an effective way of 'holding' them:

> Intervener (2.13): It was quite a random conversation … I don't like silence so I just filled it with things that popped into my head.

> Survivor (1.05): He just started rambling on about his own life, which was kind of lovely. Like he was just keeping me listening even if I wasn't talking … I could still relate to him in a way. He was saying he'd just come back from a party or something…

Silences and just being present: Other interveners recognised that it was possible to connect with a person without words and that silence could provide the suicidal person with a welcome respite from the intrusive and negative babble that was going on inside their own head:

> Survivor (1.03): She didn't really talk too much. She was just there … And she stayed with me and bought me another cup of coffee … and eventually the voices were stopping and I felt like I was in the moment.

The mere presence of another person was also a powerful deterrent. Survivors consistently reported that the arrival of another person made it impossible to go through with their planned suicidal act. Even at the point of crisis, they still seemed able to think of others:

> Survivor (1.08): I wanted them to just go away, but they didn't… They were there and I didn't want to do anything in front of their little baby … because he was old enough to know what had happened.

Embodied communication or body language was as powerful as words in establishing a non-threatening presence and conveying warmth and empathy, and included the following elements.

Proximity and positioning of self: Interveners described how they positioned themselves vis-à-vis the suicidal person, placing themselves on an equal plane (eg, sitting or squatting) and in the same orientation (alongside and facing the same way) so as to appear friendly rather than confrontational. Maintaining a respectful distance was seen as important by those who were suicidal:

> Survivor (1.09): I remember saying, Don't come so close. I was really kind of adamant about that because this was my thing and my space.

Caring gestures and touch: Interveners attended to the suicidal person's physical needs, offering them a coat, a cigarette or a warm drink. Participants in both groups recognised that the warm touch of another human being could be very powerful in breaking the isolation of the suicidal person, but several survivors reported that histories of abuse had left them with a horror of physical contact and stressed the importance of asking permission before touching:

> Survivor (1.03): She said, Do you mind if I touch your hand? She asked me, and I can remember feeling the touch of a human person so clearly.

> Survivor (1.06): Her hand was on my arm and it was warm … I was this frozen ice person, who had given up on everything, and she'd come over with sort of warm words and this physically warm touch that I didn't necessarily want to pull away from.

General demeanour: Survivors consistently reported that the manner in which the intervener spoke to them was as important as the words spoken, if not more so. Calmness (not showing alarm), authenticity (just being themselves, being 'real'), steadfastness ('not budging') and sincerity were traits that were mentioned again and again and that had the effect of making the person feel safe, valued and connected. The mere act of stopping and saying something gave a powerful signal to the person, at their lowest point, that they mattered:

Survivor (1.07): The fact that you do stop makes all the difference, rather than walking past … The fact that [on three separate occasions] people stayed and talked to me … It was like, right now you matter and your life matters and we care what happens to you. And that was a genuine thing.

### 2) Moving to a safer location

Nearly all interveners had to work out how to move the person away from the means of suicide to a safer location. Efforts to do so took one of the following three forms.

Forcible restraint: Some interventions began with a physical act of 'grabbing', designed to effect the swift and forcible removal of a person from a situation of extreme danger, before any attempt was made to engage in verbal rescue work. While no one regarded this as ideal, some survivors recognised that it had been the right thing to do in the circumstances:

Survivor (1.01): I remember hearing two people coming up the steps … and one of them just pulled me backwards off the ledge … Had I not been pulled down, I doubt I would have engaged at all because I was so inside of myself.

The risks involved in grabbing were also clearly illustrated. In one case, two women walking on a riverside path saw a person on a bridge ahead of them, on the wrong side of the railing, leaning out over the water and in extreme peril. Having reached the person in time and grabbed both her arms, they then found themselves trapped, lacking the strength to pull her back over the railing, struggling to keep hold, and able neither to engage her in conversation nor to attract the attention of passing motorists. The situation was eventually resolved, but not without injury to one of the interveners.

Grabbing was also seen as a purely instrumental action, which, while it may have been effective in saving the life on that occasion, did nothing to contribute to longer term recovery. Interventions at the other end of the control/agency spectrum (figure 2), which encouraged self-rescue, were deemed more likely to have an ongoing positive and protective effect.

Direct appeal ('Please come away from the edge'): This was an option when there was less urgency and it was believed that the person had the capacity to respond. Some approached it in an oblique manner:

Intervener (2.21): I didn't want to confront her and say, Don't jump! … I said, Do you realise the cliff edge is really crumbly there? I'm quite worried that it might collapse. Why don't you step back a bit?

Indirect ('Let's go somewhere for a coffee'): Offering to buy the person a warm drink and something to eat performed multiple purposes. It could be both a simple act of kindness and validation, and also a ploy to move the person to a safer location. Either way, it was a highly effective action, serving also to ease the person back into the normal, everyday world (figure 1):

Intervener (2.12): I was trying to work out how I could get her out of the tube [underground rail network] and somewhere safer … So I said, Let's go and get a cup of tea somewhere and have some cookies … And she helped herself to a cookie and there were signs she was coming to a little bit … and she sort of opened up to me then.

### 3) Summoning help

There were three potential sources of support on which interveners could call: a family member or friend nominated by the suicidal person; other passers-by, and the emergency services.

Family member or friend: Interveners typically asked about family and friends as part of 'bursting the bubble,' but it was also a way to identify someone who might come and take the person home. The following example shows how a brief verbal exchange performed both tasks at once. The intervener appears initially to be trying to establish if there is someone who can be called to provide support. The suicidal person can think of no-one, but as the exchange proceeds she gradually remembers her extended family and the closeness of her social bonds (figure 1). The sight of the photograph on her phone represented a turning point, after which she could no longer go through with the act of suicide and agreed to contact one of her daughters:

Survivor (1.03): She said, Is there somebody you can ring? And I said, No, there's nobody. And she said, Have you got any children? And I said, Yes, I've got two daughters. Oh, have you got a picture? she said. And I got out my phone and showed her a picture of my six grandchildren…

Other passers-by: Efforts to enlist help from other passers-by were rarely successful. Participants in both groups gave accounts of busy locations and of people passing by, either not noticing or pretending not to notice what was happening. Several interveners made repeated and desperate attempts to signal to other passers-by but were ignored, although in some cases it transpired that a member of the passing crowd had taken it on themselves to call emergency services. However, other interveners rejected offers of help, fearing that the presence of another person would threaten the one-to-one connection and trust they were establishing with the suicidal person.

Emergency services: Interventions typically ended with the arrival of emergency services. Rail staff and highways officers had usually alerted emergency services prior to approaching the suicidal person, so that their task was simply to keep the person safe until specialist help arrived. For members of the public, it was much more challenging. None of them had thought to call emergency services prior to intervening, and once they had approached and

embarked on rescue work it became very difficult to do so. They feared that any move to do so, in particular, any mention of police involvement, would trigger panic and flight. In some cases, the intervener asked permission to call for help and it was refused, leaving them alone and floundering. Others resorted to subterfuge or were fortunate that a member of the crowd called for help on their behalf, often unbeknown to them. Several interveners reported being unsure which service to ask for. The following example illustrates the multitasking nature of intervention work, the fragility of the connection with the suicidal person, and the impossibility of maintaining it while doing other things:

> Intervener (2.12): I was trying to get my phone out of my handbag without losing eye contact … And I glanced down to unlock it and she ran … In that split second I lost her … So I was going after her and I was calling 999 as well, and I remember on the phone feeling quite confused because I really wasn't sure what service to be asking for … I said, There's a lady running into the road, and they said, Well, what service do you need? And I said, I don't know…

Interveners who had worked hard to build an emotional connection with the person expressed misgivings about medicalising the situation by calling an ambulance or criminalising it by involving the police.

### Endings and aftermath

Ending an intervention without a handover to emergency services or to a family member or friend was fraught with difficulty. Interveners described being desperate to extricate themselves, sometimes after several hours, but not knowing whether it was safe to let the person to go on their way and sometimes debating whether they should invite the person home with them.

Endings involving a handover were not necessarily any easier. Interveners experienced intense relief as they themselves were 'rescued', but were often left with feelings of exclusion, loss and fear of consequences for the person (especially when they were taken away in handcuffs or sectioned under the Mental Health Act). Some were left feeling disturbed by what had happened. Those who had been on their way to work at the time described the surrealness of trying to carry on as normal after the extraordinariness and intensity of the encounter.

### DISCUSSION

This is the first empirical study to examine the role of passing strangers in preventing suicides. Intervention was seen to involve three main tasks: 'bursting the bubble' (reconnecting with self, others and everyday world); moving to a safer location and summoning help. Interveners accomplished these tasks in a range of ways, using both verbal and non-verbal communication and different degrees of restraint.

Those who had been suicidal (the survivor group) typically described themselves as being dissociated or 'in a bubble,' cut-off from themselves and everything around them. Dissociative states have long been linked with suicide. Orbach, for example, postulates that detachment from the body, indifference to physical pain and lack of affect are necessary in order for an individual to go through with a suicidal act.[19] Absence of emotion is a subtle and counterintuitive clue. In this regard, suicide is like drowning insofar as the person does not look like we expect them to. A drowning person is not thrashing about and shouting for help, as commonly portrayed in film; they are silent and still and are easily overlooked. Likewise the suicidal person rarely conforms to the stereotypical image of the anguished soul, as depicted in Munch's painting of The Scream. This is an important message to convey to the public. In an online survey of rail staff, Mackenzie *et al* found that a high proportion of staff expected suicidal individuals to exhibit visible signs of distress,[20] suggesting that this may be a common misperception.

With few outward clues except location, interveners relied on their own 'sixth sense' that something was wrong. Some quality of oddness or incongruity triggered an alarm inside them, which they were able to hear. Thus recognition relies as much on an internal state of openness, curiosity, attunement to others and attention to one's own inner voice as on any external signs.

Openness needs to be combined with readiness for action. The interveners in our study were ordinary people; they did not possess any special knowledge or skills. What was unusual about them was their lack of hesitancy and their disregard of the social risks involved in approaching a stranger in a public place. Most people are kept from doing so by fear of a rude rebuff and resulting embarrassment.[21] The interveners we interviewed were not fearless but, on the principle of 'Feel the fear and do it anyway', they did it anyway. They also showed themselves to be resourceful, able to dig around in their personal 'experience bag' and find things that worked, without necessarily knowing how or why.

Our findings suggest that people do not need to be provided with an intervention script, nor should they be afraid of saying 'the wrong thing'. The words the intervener spoke were not nearly as important as how they made the person feel. Survivors described interveners as having broken through the dissociative state, dispelling feelings of fear, isolation and worthlessness and making them feel safe, validated and reconnected. The basic mechanisms are depicted in figure 3.

This model of effective intervention, with feeling rather than reasoning at its core, runs counter to conventional wisdom, as contained in programmes such as the Applied Suicide Intervention Skills Training (ASIST)[22] and in academic literature.[8] The ASIST model stresses the importance of listening closely to the person's reasons for dying and working with them to uncover reasons for living. Its focus is on verbal reasoning. Likewise, Omer

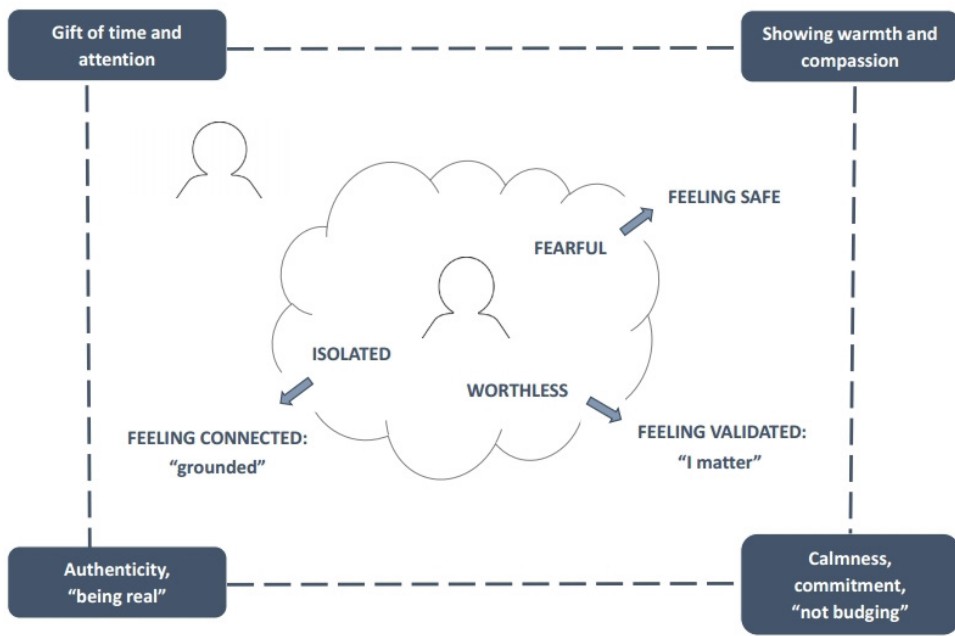

**Figure 3** Proposed mechanisms of action in an effective intervention by a stranger in a public location.

and Elitzur's proposed script for a last-minute crisis intervention insists on challenging suicidal thinking. The authors compare the situation to a court of law in which the defence lawyer (the intervener) picks holes in the prosecutor's (the suicidal person's) story in an attempt to win the argument.[8] Our interveners were not attempting to win a legal battle. Nor were they trying to imitate therapists or crisis negotiators. They were just being themselves and reaching out as one human being to another. This is not to say that reasoning and verbal persuasion played no part, or that they may not be important at other stages of the suicidal process, but few of our survivors reported having been swayed by logic or argument. What they recalled most strongly about those who had helped them was their authenticity and their compassion. What made the difference was the simple fact that someone cared enough not to walk on by.

Our model may not be appropriate to all suicide crisis settings. A situation involving an unknown person on the point of carrying out a suicide plan in a public location is very different from other settings. These include domestic or social settings, where a personal relationship exists between the two parties;[15] clinical settings, where a professional relationship and statutory responsibility for risk management exist,[23–25] and telephone counselling services, where the suicidal person has invited a talking intervention by calling a crisis line.[26 27] In these situations, the person may not yet be at the stage having formulated a suicide plan. Our study suggests that rather than there being a 'one size fits all' model of human intervention, as the ASIST training supposes, different social contexts and stages of the suicidal process may require different approaches.

Our findings show clearly that anyone who wants to intervene can safely do so. Last-minute intervention requires no specific learning and can be highly effective when spontaneous and unscripted. In 2018, Network Rail in partnership with Samaritans launched a campaign entitled 'Small Talk Saves Lives', encouraging rail travellers simply to say 'Hello' and strike up a normal conversation if they are concerned about someone.[28] Our data confirm that this campaign message is entirely appropriate as far as the initial approach to a vulnerable person is concerned. However, it fails to recognise the intense, prolonged and taxing nature of intervention, the complex juggling acts that interveners may have to perform in the course of trying to keep someone safe, and the troubling emotions they may be left with. It is no small thing to save a life. The conundrum for public education is how to prepare people adequately for the challenges they may face without deterring them from intervening.

A set of simple public education messages is needed to encourage people to recognise and reach out to those who may be at risk of suicide in public locations. We are working with a wide range of non-academic partners to develop and disseminate these.

**Acknowledgements** We thank the participants in both groups for sharing their stories. We are also grateful to Network Rail and Highways England for help with recruiting interveners, to Jonny Benjamin MBE, Neil Laybourn (the original 'Stranger on the Bridge') and the other members of our expert advisory group.

**Contributors** CO had the idea for the study and conducted public involvement. She designed the study with CA, managed the project, led the analysis of data and wrote the manuscript. JD collected data and analysed the transcripts. Both JD and CA commented on emerging drafts of the manuscript. All authors have read and approved the final version. CO is the guarantor.

**Funding** This work was supported by the Medical Research Council, grant number MR/P01707X/1.

**Competing interests** None declared.

**Patient consent for publication** Not required.

## Open access

**Ethics approval** Ethical approval was granted by the University of Exeter Medical School Research EthicsCommittee.

**Provenance and peer review** Not commissioned; externally peer reviewed.

**Data availability statement** Reasonable requests for data will be considered but will be subject to additional ethical approval.

**ORCID iD**
Christabel Owens http://orcid.org/0000-0001-9846-0889

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
