## [Reviewer comments · BMJ Open]

ARTICLE DETAILS

TITLE (PROVISIONAL)	Intervening to prevent a suicide in a public place: A qualitative study of effective interventions by lay people
AUTHORS	Owens, Christabel; Derges, Jane; Abraham, Charles

VERSION 1 – REVIEW

REVIEWER	Michael Kral Wayne State University United States
REVIEW RETURNED	19-Jun-2019

GENERAL COMMENTS	This is an excellent study on public prevention of suicide. It is unique and very well written. I just have a few suggestions: 1. I would lengthen the abstract. 2. On p. 8 you identify interview guides used. I recommend listing the "topics of interest" that were queried.
---

REVIEWER	sofian Berrouguet brest university hospital
REVIEW RETURNED	11-Jul-2019

GENERAL COMMENTS	Autors provided and interesting report of a qualitative study conduced in a suicide attemptes and suicide "rescuers" population. No innovative method was poposed, but the choic of the population is critical and relevant. The results are clearly presented, and the "bubble" image of the brings an relevant clinical illustration of 1."The aim of this study was to identify the core components of an effective intervention by a member of the public in a 'Stranger on the Bridge' situation." i suggest to replace the term stanger on the brige which sounds funny or "film title". The objectives and related outcomes measures should be clearly described in the method section 3. Discussion section starts by the statment "It demonstrates that anyone can prevent a suicide in a public place, and provides clear evidence-based principles to guide potential interveners". Authors should start with a more precise statement , for example summerizing their principal results 4. The title and overall writing does not reflects enough the general "recommandation" tone of the paper. Again, the results are very
--

	interesting, and authors should highlight the core results, which to my opinion are the guidelines to "rescue a suicide attempter".
--	---

VERSION 1 – AUTHOR RESPONSE

Reviewer 1:

This is an excellent study on public prevention of suicide. It is unique and very well written. I just have a few suggestions:

1. I would lengthen the abstract.

Our response: The stated word limit is 300 words and our submitted abstract was carefully crafted so as to be within the limit. It is not clear what additional information Reviewer 1 thinks we should include, but we have been guided by Reviewer 2's comments and have expanded the Conclusions in order to make the overall message clearer. The abstract is now just under 350 words, which we hope is acceptable to the journal.

2. On p. 8 you identify interview guides used. I recommend listing the "topics of interest" that were queried.

Our response: As space is limited in the article itself, we have uploaded the interview schedules as supplementary files, and have indicated this in the text.

Reviewer 2:

Authors provided an interesting report of a qualitative study conducted in a suicide attempters and suicide "rescuers" population. No innovative method was proposed, but the choice of the population is critical and relevant. The results are clearly presented, and the "bubble" image of the brings an relevant clinical illustration of

1. "The aim of this study was to identify the core components of an effective intervention by a member of the public in a 'Stranger on the Bridge' situation." I suggest to replace the term stranger on the bridge which sounds funny or "film title".

Our response: We accept this point. We used the film title as a convenient shorthand to save a lengthy description each time, but we have changed all instances, except when it actually refers to the title of the film (as in Introduction paragraph 1 and the Acknowledgements).

2. The objectives and related outcomes measures should be clearly described in the method section.

Our response: In our view, aims/objectives are not part of methods but should precede them. Our aims are stated at the end of the introduction, in keeping with usual practice.

This was a qualitative study not a trial, so no outcomes were measured. Our interviews were narrative-based and their broad scope is described in the manuscript. Readers can also refer to the interview schedules for full details of the topics covered.

3. Discussion section starts by the statement "It demonstrates that anyone can prevent a suicide in a public place, and provides clear evidence-based principles to guide potential interveners". Authors should start with a more precise statement, for example summarizing their principal results.

Our response: We have amended the first paragraph of Discussion so that it now presents our principle findings (mirroring what is in the Abstract). The conclusions we draw from them follow.

4. The title and overall writing does not reflect enough the general "recommendation" tone of the paper. Again, the results are very interesting, and authors should highlight the core results, which to my opinion are the guidelines to "rescue a suicide attempter".

Our response: We have revised the title to make it shorter and more sharply focused on intervention. We are unable to include results in the title, as this would contravene journal policy. We believe that the core results are sufficiently clear in the revised abstract and discussion, and in the figures.

VERSION 2 – REVIEW

REVIEWER	Michael Kral Wayne State University, Detroit, MI, USA
REVIEW RETURNED	19-Sep-2019

GENERAL COMMENTS	This is an excellent paper on lay people preventing a public suicide. I have only a few comments/questions. 1. Who conducted the interviews? 2. On p. 9 you indicate that the question was designed in consultation. Were these the interview questions?
--

VERSION 2 – AUTHOR RESPONSE

Reviewer: 1

1. Who conducted the interviews?

Our response:

The interviewer was already identified by her initials [JD] on p.8 of our manuscript.

2. On p. 9 you indicate that the question was designed in consultation. Were these the interview questions?

Our response:

Apologies for lack of clarity. We meant that the research question was formulated in consultation with people with lived experience and other stakeholders. To avoid adding more words, we have removed this so that the sentence now says simply, "We designed the study in consultation..."